# A Simple and Effective Visual Fluorescent Sensing Paper-Based Chip for the Ultrasensitive Detection of Mercury Ions in Environmental Water

**DOI:** 10.3390/s23063094

**Published:** 2023-03-14

**Authors:** Jinglong Han, Huajun Liu, Ji Qi, Jiawen Xiang, Longwen Fu, Xiyan Sun, Liyan Wang, Xiaoyan Wang, Bowei Li, Lingxin Chen

**Affiliations:** 1School of Environment and Materials Engineering, Yantai University, Yantai 264005, China; 2CAS Key Laboratory of Coastal Environmental Processes and Ecological Remediation, Research Center for Coastal Environmental Engineering and Technology, Yantai Institute of Coastal Zone Research, Chinese Academy of Sciences, Yantai 264003, China; 3Center for Ocean Mega-Science, Chinese Academy of Sciences, Qingdao 264003, China

**Keywords:** CdTe quantum dots, mercury ions, paper-based chips, silica nanospheres, rapid detection

## Abstract

Traces of mercury ions in environmental water can harm humans and animals. Paper-based visual detection methods have been widely developed for the rapid detection of mercury ions; however, existing methods are not sensitive enough to be used in real environments. Here, we developed a novel, simple and effective visual fluorescent sensing paper-based chip for the ultrasensitive detection of mercury ions in environmental water. CdTe-quantum-dots-modified silica nanospheres were firmly absorbed by and anchored to the fiber interspaces on the paper’s surface to effectively avoid the unevenness caused by liquid evaporation. The fluorescence of quantum dots emitted at 525 nm can be selectively and efficiently quenched with mercury ions, and the ultrasensitive visual fluorescence sensing results attained using this principle can be captured using a smartphone camera. This method has a detection limit of 2.83 µg/L and a fast response time (90 s). We successfully achieved the trace spiking detection of seawater (from three regions), lake water, river water and tap water with recoveries in the range of 96.8–105.4% using this method. This method is effective, low-cost, user-friendly and has good prospects for commercial application. Additionally, the work is expected to be utilized in the automated big data collection of large numbers of environmental samples.

## 1. Introduction

Trace amounts of mercury ions in the environment can cause harm to humans and animals [1,2,3]. Normally, sample storage, transportation and pretreatment are necessary when using traditional instrumentation methods for the detection of mercury ions at trace levels, which are not adaptable to the needs of rapid on-site environmental analysis [4,5,6]. In contrast, paper-based visual analytical methods have been developed for rapid on-site environmental analysis due to their being user-friendly, disposable and low-cost [7,8,9]. To date, molecular and nano-materials have been widely developed for the construction of visual sensing methods on paper-based devices for the rapid detection of mercury ions, such as organic chromophores/fluorophores [10], plasma nanoparticles [11], quantum dots [12], carbon dots [13] and so on. Yang et al. developed a paper-based chip based on the catalytic effect of the combination of Hg^2+^ and gold nanospheres on the colorimetric reaction of tetramethyl benzidine and H_2_O_2_ for the detection of mercury ions [14]. Monisha et al. reported silver nanoparticle-based colorimetric sensing on a paper-based chip combined with the use of smartphones for the quantitative detection of mercury ions [11]. Patil et al. developed a novel rhodamine-based colorimetric/fluorescent chemical sensor for the detection of mercury ions [15]. Kitchawengkul et al. combined nitrogen-doped carbon dots with μPAD for the colorimetric determination of the total cholesterol in human blood [16]. Li et al. found that mercury ions could promote the oxidation effect of CdSe/ZnS quantum dots catalyzed using 3,3,5,5-tetramethylbenzidine under visible light, with a detection limit of 0.09 µM for mercury ions [12]. All of these reported methods have successfully achieved the visual detection of mercury ions in water samples. However, most of the existing visual colorimetric methods have poor sensitivity, poor stability and are time-consuming, and cannot meet the need for the rapid and sensitive on-site detection of trace mercury ions in real environmental samples.

Nanomaterial-based fluorescent sensing is characterized by high sensitivity and selectivity [17,18,19]. Furthermore, fluorescence emission in the visible wavelength of most fluorescent nanomaterials offers the possibility of their application in visual sensing [20,21,22]. Among them, quantum dots (QDs), as a nanomaterial with high fluorescence yield, are widely used in bioimaging, biological and chemical sensors [23,24,25]. The construction of nanomaterial-based fluorescent sensing on paper-based chips can enhance the user-friendliness of the sensing process, reduce sample volume and save costs. Related studies have been widely reported [19,26,27]. In our previous work, we grafted QDs and a molecularly imprinted polymer onto a rotating paper-based microfluidic chip for the rapid and multiplex fluorescent detection of 4-nitrophenol and 2,4,6-trinitrophenol in real environments [22]. It is worth exploring how to develop highly sensitive paper-based visual fluorescent sensing methods using QDs. Chen et al. grafted nano-ZnTPyP and QDs onto paper-based chips for the rapid detection of caffeine in food and biological samples [28]. Due to the intricate and random characteristics of paper fibers, the concentrated and uniform distribution and solidity of quantum dots on paper fibers are the keys to their excellent visual sensing effect. Anchoring of QDs on paper fibers using chemical grafting has been reported, but the chemical bonding process introduces multiple reagents and is tedious. In addition, the chemical grafting of quantum dots formed on the surface of paper fibers cannot avoid local grafting defects caused by the randomness of paper fibers [29,30]. All of these factors directly affect the sensitivity and reliability of QDs–paper-based visual fluorescent sensing.

To overcome the problems associated with the use of chemically grafted quantum dots on paper fibers for the sensitive and reliable visual fluorescent detection of mercury ions, we developed a simple and effective visual fluorescent sensing paper-based chip. CdTe-quantum-dot-modified silica nanoparticles were firmly absorbed by and anchored to the fiber interspaces of the paper’s surface, effectively avoiding the unevenness caused by liquid evaporation. The quantum dot fluorescence emission was quenched by trace mercury ions and the results could be visualized on a paper-based chip. Under UV light, the color signal of fluorescence on a paper-based chip was captured using a smartphone to generate grayscale data for the rapid detection of mercury ions in environmental water. This method is effective, low-cost, user-friendly and has good prospects for commercial application.

## 2. Experimental Methods

### 2.1. Main Reagents

Tellurium, sodium borohydride, cadmium chloride hemi(pentahydrate) (CdCl_2_·2.5H_2_O), thioglycolic acid, (3-aminopropyl) triethoxysilane (APTES), 1-(3-dimethylaminopropyl)-3-ethylcarbodiimide hydrochloride (EDC), N-hydroxy succinimide (NHS), mercury chloride (HgCl_2_), etc., were ordered from Aladdin (Shanghai, China) Reagent. Ethanol absolute, NaOH, ammonia solution, tetraethyl orthosilicate (TEOS), KCl, CaCl_2_, NaCl, Mg(NO₃)₂, AlCl_3_, Zn(NO_3_)_2_·6H_2_O, FeCl_3_·6H_2_O, AgNO_3_, CoCl_2_, CuCl_2_, Ni(NO₃)₂·6H₂O, BaCl_2_·2H_2_O, CrCl_3_·6H_2_O and other accessory materials were ordered from Sinopharm Chemical (Shanghai, China) Reagent. All solvents, chemicals and materials were at least of analytical purity. Whatman No. 1 chromatography paper was purchased from GE (Shanghai, China) for further sizing. Ultrapure water (specific resistance 18.2 MΩ) was produced by Pall Cascada Laboratory Water Systems (Millipore, Bedford, MA, USA). Paper-based chips were printed using a Xerox Phaser wax spray printer, and the paper and material were characterized using a scanning microscope (SEM; JSM 5600 L V). Fluorescence intensity was measured using a fluorescence spectrophotometer (Hitachi F-7000) with a quantum dot excitation wavelength of 396 nm and an emission wavelength of 450–650 nm, using a UV spectrophotometer (NanoDrop 2000/2000 C, Thermo Scientific) to collect UV-visible spectra. The collected images were processed and grayscale values were detected using ImageJ software. The model of the smartphone used in the visual inspection was Huawei VOG-AL10.

### 2.2. Preparation of the Paper-Based Chip

The pattern of the paper-based chip was designed by Adobe Illustrator software. The diameter of the circle hydrophilic sensing area was 6 mm, and the rest of the black parts were hydrophobic area. Whatman No. 1 filter paper was cut to the size of A4 paper, and the pattern was printed onto the filter paper using a Fuji Xerox wax jet printer at the maximum resolution of 2400 dpi. The printed filter paper was placed face up at 150 °C. The black wax was melted and penetrated downward by gravity to form a hydrophobic barrier on the filter paper.

### 2.3. Synthesis of Thioglycolic-Acid-Modified CdTe QDs

Tellurium powder and sodium borohydride were mixed in an ethanol/water solution (*v*:*v* 3:1). NaHTe supernatant was obtained after heating at 40 °C for 4 h. A total of 68.4 mg of CdCl_2_ and 63 μL of thioglycolic acid were dissolved in 75 mL water, and the pH of the solution was adjusted to 9 with 1 mol/L NaOH. Then, 1 mL of NaHTe was added. Orange CdTe QDs with an emission wavelength at 525 nm were obtained under nitrogen protection and heated at 95 °C for 30 min.

### 2.4. Synthesis of Amino-Modified Silica Nanospheres

Amounts of 30 mL of ethanol, 50 mL of water and 10 mL of ammonia solution were mixed together, then 25 mL of a TEOS/ethanol (*v*:*v*, 1:4) mixture was added dropwise for 6 h of reaction. Then, 5 mL of APTES was added for 12 h of reaction. The product was dried after washing three times with water.

### 2.5. Synthesis of QD-Grafted Silicon Nanospheres (QDs–SiO_2_)

Amounts of 5 mL of QDs and 80 mg of silica spheres were dissolved in 15 mL of water. Then 3 mL of EDC and 3 mL of NHS were added. After stirring for 30 min at room temperature, the QDs–SiO_2_ was washed three times with water, and 5 mL of water was added from reserve.

### 2.6. Preparation of QDs–SiO_2_ Paper-Based Chips and Detection of Mercury Ions

Next, 5 μL of QDs–SiO_2_ solution was added to the reaction area on paper-based chips and dried at 40 °C. The dried paper-based chips were stored at 4 °C away from light. Using the paper-based chip was very simple. The 5 μL sample containing mercury ions was added to the reaction area and, after waiting for 90 s at room temperature, the color signal of green fluorescence on the paper-based chip was collected using a smartphone camera in the UV light (365 nm) box, which had a UV lamp power of 6 w. The smartphone was located 25 cm above the paper-based chip and stayed fixed, the flash was turned off and the image was in the center of the smartphone. The smartphone captured photos using the auto-focus mode. After capturing the photos, the grayscale value of the image’s reaction area was measured using ImageJ software, and a calibration curve was established. In addition, the fluorescence signal was obtained using a fluorescence spectrophotometer for the cooperative verification of the test results from the smartphone. A diagram of the manufacturing and testing of paper-based chips is shown in Figure 1.

## 3. Results and Discussion

### 3.1. Sensing Mechanism and Analytical Process of the Paper-Based Chip

In order to enhance the uniform distribution and stable anchoring of QDs concentrated on the surface of paper fibers, a simple and effective method has been proposed. We grafted QDs onto the surface of silica nanospheres to form relatively large-volume composites (QDs–SiO_2_). Owing to the electrostatic adsorption and trapping of the QDs–SiO_2_ by the surface of the paper fibers and their interspaces, the QDs–SiO_2_ could be anchored centrally onto the surface of the paper fibers. The quantum dots attached to the surface of the silica nanospheres retained their sensing properties. The fluorescence of CdTe QDs was quenched by Hg^2+^ ions, which resulted from the electron transfer from QDs to Hg^2+^ [31]. The fluorescence intensity of the QDs gradually decreased as the concentration of Hg^2+^ ions increased, thus causing the faded green color in the images of the paper-based chip taken using a smartphone. The gray values of the green color in the images were analyzed using software to command the concentration of Hg^2+^ ions.

### 3.2. Microstructure Characterization

The amino-modified SiO_2_ nanospheres, QDs–SiO_2_, bare paper and QDs–SiO_2_ paper were characterized using scanning electron microscopy (SEM). As shown in Figure 2A, the surface of the amino-modified SiO_2_ nanospheres was relatively smooth and the particle size distribution was between 500 and 1000 nm. Figure 2B shows that there were dense particles on the surface of the SiO_2_ nanospheres, which proved that the QDs were successfully grafted onto the surface of the SiO_2_ nanospheres. Figure 2C shows a heterogeneous, dendritic fiber porous structure. Figure 2D shows that QDs–SiO_2_ was attached to the surface of the fiber structure and the paper fiber interspaces were filled with QDs–SiO_2_. At high magnification (insert image of Figure 2D), the concentrated state of QDs–SiO_2_ could be observed, which proved that QDs–SiO_2_ was successfully anchored onto the surface of the paper fibers.

### 3.3. Investigation of Sensing Conditions

In order to improve the accuracy and stability of the sensing system, we optimized important preparation and testing conditions.

The amounts of QDs and SiO_2_ nanospheres used in the preparation of QDs–SiO_2_ were investigated. Adequate grafting of quantum dots onto the SiO_2_ nanospheres’ surface is important to obtain efficient sensing on paper-based chips. We optimized the ratio of silicon spheres to QDs by detecting the fluorescence intensity of the supernatant. A_0_ indicates the initial fluorescence intensity of the QDs. A indicates the fluorescence intensity of the QDs (excess QDs in the supernatant) that failed to graft onto the surface of the SiO_2_ nanospheres during the reaction. (*A_0_ − A*)/*A_0_∗100%* denotes the utilization of a determined amount of QDs. As shown in Figure 3A, as the amount of silica spheres kept increasing (40–120 mg), the utilization of QDs (20 mL) increased at first and then gradually stabilized. Finally, 20 mL of QDs and 80 mg of SiO_2_ nanospheres were selected as the optimal conditions. In addition, the reaction time for grafting QDs onto silica nanospheres is also an important condition. As shown in Figure 3B, the utilization of QDs increased significantly within the time range from 0 to 20 min, and then stabilized when the grafting time exceeded 20 min. Therefore, 20 min was chosen as the optimized reaction time.

The buffer system may have an impact on sensing. We investigated three buffer systems (PBS, Tris and HEPES, pH 7.5) on the paper-based chip. The effect of sensing can be indicated by (*G_0_/G*) − 1. G_0_ is the gray value of the sensing area image on absent Hg^2+^ ions. G is the gray value on 50 μg/L of Hg^2+^ ions. As shown in Appendix A, the sensing effect was almost lost in the Tris buffer system, the reason for which may be that the Tris buffer has a certain quenching effect on QDs. Comparing the PBS and HEPES buffer systems, the effect of sensing was better in the HEPES buffer system, so we chose HEPES as the buffer system. Furthermore, we found that the effect of sensing was best at a pH of 7.5 (Figure 3C); therefore, a pH of 7.5 was selected for the subsequent detection.

The equilibration time of the sample on the paper-based chip was also investigated. The QDs–SiO_2_ on the paper-based chip in dry condition was maintained in a low-fluorescence-intensity state. After the addition of the sample, the fluorescence intensity of the QDs–SiO_2_ recovered over time, while being affected by the quenching of Hg^2+^ ions in the sample. Eventually, after a period of time, the fluorescence intensity of QDs–SiO_2_ reached equilibrium. In order to determine an optimal reaction time, we examined the change in the gray value of the paper-based chip after the addition of the sample (containing 50 μg/L of Hg^2+^ ions). As shown in Figure 3D, the gray value was stabilized after the sample dripped for 90 s. This paper-based chip assay allows the signal to be collected only 90 s after introducing the sample.

### 3.4. Analytical Performance of the Paper-Based Chip

Under optimal conditions, we studied the efficiency and sensitivity of the paper-based chip. After capturing the images’ sensing area via smartphone, the gray values of the images were extracted using Image J software. The gray values decreased with the increasing concentration of Hg^2+^ ions. Figure 4A shows that the reduction in grey scale values was positively related to the concentration of Hg^2+^ ions in the detection range of 0–100 μg/L, with a linear regression equation y = 0.216x + 0.686, R^2^ = 0.993, and the limit of detection (LOD) was 2.83 μg/L (the formula is 3*σ/m, σ is the standard deviation of the gray value of the blank group and m is the slope of the calibration curve).

To further verify the visual sensing effects on paper-based chips, we investigated the detection performance of Hg^2+^ ions on paper-based chips using a fluorescence spectrophotometer. As shown in Figure 4B, the fluorescence intensity decreased with increasing mercury ion concentration, the linear regression equation in the detection range of 0–100 μg/L was y = 0.0086x + 0.0576, R^2^ = 0.984 and LOD was 0.38 μg/L. Compared to visual sensing methods, fluorescence spectroscopy was more sensitive. Certainly, a high-precision fluorescence spectrometer was required.

To highlight the advantages of the uniformity and stability of QDs–SiO_2_ on paper-based chips, we investigated the behavior of QDs on paper-based chips. As shown in Figure 4C, the QDs–SiO_2_ formed a uniform green spot and the QDs appeared obviously non-uniform on the paper-based chip under UV light. Several nanometer-sized (<10 nm) quantum dots dispersed in solution could move with the flow of the liquid in the paper fiber. The coffee ring effect of QDs was caused by the local non-uniformity of the liquid evaporating from the paper-based chip [32,33]. These issues caused some errors (the relative standard deviation (RSD) of the gray value was 2.3%) in data acquisition. Meanwhile, QDs–SiO_2_ exhibited uniform and stable activity on the paper-based chip, and the RSD of the gray value was 0.35%. Therefore, our method was very effective.

### 3.5. Investigation of Selectivity

The selectivity of the sensing method originated from the adjustment of the fluorescence emission wavelength of the QDs. As shown in Figure 4D, the response of different emission wavelengths of QDs for the detection of Hg^2+^ ions differed. The QDs with a fluorescence emission wavelength of 525 nm had the best sensing effect for the detection of Hg^2+^ ions.

To investigate the selectivity of the sensing method, we selected a variety of interferents at 50 μg/L, including potassium ions (K^+^), calcium ions (Ca^2+^), sodium ions (Na^+^), magnesium ions (Mg^2+^), aluminum ions (Al^3+^), zinc ions (Zn^2+^), iron ions (Fe^3+^), silver ions (Ag^+^), cobalt ions (Co^2+^), copper ions (Cu^2+^), nickel ions (Ni^2+^), barium ions (Ba^2+^), chromium ions (Cr^3+^), cadmium ions (Cd^2+^) and mercury ions (Hg^2+^). As shown in Appendix A, the impact of interferents on sensing was within acceptable limits, which proved that QDs–SiO_2_ had good selectivity.

### 3.6. Practical Applications

In order to verify the practical performance of the paper-based chip, we used the sensor to detect Hg^2+^ ions in real water samples. Three seawater samples were taken from three different sites along the Yellow Sea. The lake water sample was taken from Sanyuan Lake at Yantai University. The river water sample was taken from the Guangdang River in Yantai. The tap water sample was taken from the Laishan district of Yantai. The samples were pretreated by running them through a 0.22 μm microporous filter membrane three times. The results showed that the concentration of Hg^2+^ ions in all water samples was below the detection limits of our method. Hg^2+^ ions were added to the water samples at concentrations of 10 μg/L, 50 μg/L and 100 μg/L, respectively. As shown in Table 1, the experiments showed that the recoveries of seawater were 98.7~104.8%, 97.8~100.3% and 103.55~105.4%, and the RSDs were 2.4~4.3%, 2.2~5% and 3.2~4%, respectively. The recoveries of lake water were 96.8~102.4%, with RSDs of 1.3~4.4%, and the recoveries of river water were 104.2~105.4%, with RSDs of 2.2~2.8%. The recoveries of tap water were 96.6~101.15%, with RSDs of 2.3~3.1%. These data proved that the performance of the paper-based chip was satisfactory in actual samples.

There were content differences between the water samples. Organic matters in environmental water samples may adsorb Hg^2+^, resulting in a reduced content of free Hg^2+^ in the water samples. For the simplification of the testing process, we did not use sample pretreatment techniques on the environmental water samples, and we were unable to completely remove organic matters. However, in the future we will improve our methodology by rectifying both of those issues.

### 3.7. Comparison of Different Methods

The performance of the paper-based chip and previously reported visual sensing methods for the determination of Hg^2+^ ions is presented in Appendix A [11,12,13,15,29,30]. Most methods are not sufficiently sensitive or stable. In addition, the time-consuming testing process does not meet the need for rapid on-site testing. In contrast, our method is both sensitive and stable, and faster than other methods.

## 4. Conclusions

We developed a novel, simple and effective visual fluorescence sensing paper-based chip, combined with a smartphone, for the ultra-sensitive and rapid detection of Hg^2+^ ions in environmental water in less than 90 s. After optimization of the reaction conditions, this method was able to achieve a limit of detection of 2.83 μg/L. Using a fluorescence spectrophotometer, the limit of detection could be reduced to 0.38 μg/L. This method was effective and had good selectivity. We successfully achieved the trace spiking detection of seawater (from three regions), lake water, river water and tap water with recoveries in the range of 96.8–105.4% using this method. To the best of our knowledge, it is the most sensitive visual analytical method for the determination of Hg^2+^ ions so far. This method is effective, low-cost, user-friendly and satisfies the detection needs for both freshwater and seawater. It has good prospects for commercial application. In addition, the methodology we have outlined is expected to be utilized for the automated high-throughput collection of a large number of environmental samples.

## Figures and Tables

**Figure 1 sensors-23-03094-f001:**
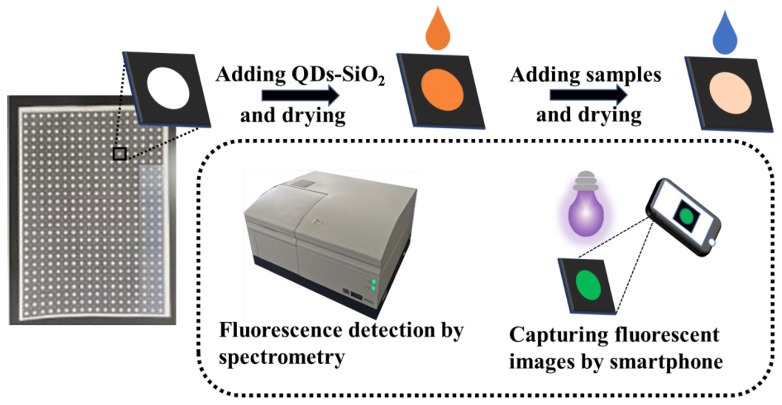
Schematic diagram of the manufacturing and testing of the paper-based chip.

**Figure 2 sensors-23-03094-f002:**
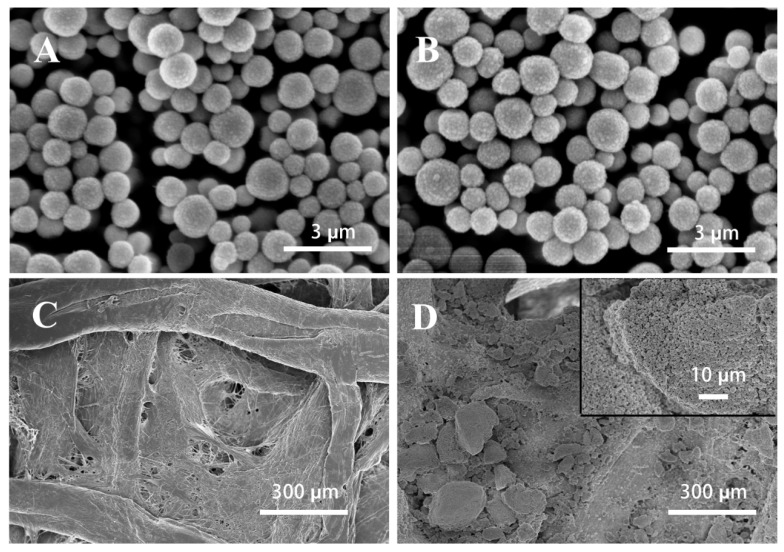
(**A**) SEM image of amino-modified SiO_2_, scale bars are 3 μm. (**B**) SEM image of QDs–SiO_2_, scale bars are 3 μm. (**C**) SEM image of Whatman No. 1 chromatography paper, scale bars are 300 μm. (**D**) SEM image of Whatman No. 1 chromatography paper titrated with QDs–SiO_2_, scale bars are 300 and 10 μm.

**Figure 3 sensors-23-03094-f003:**
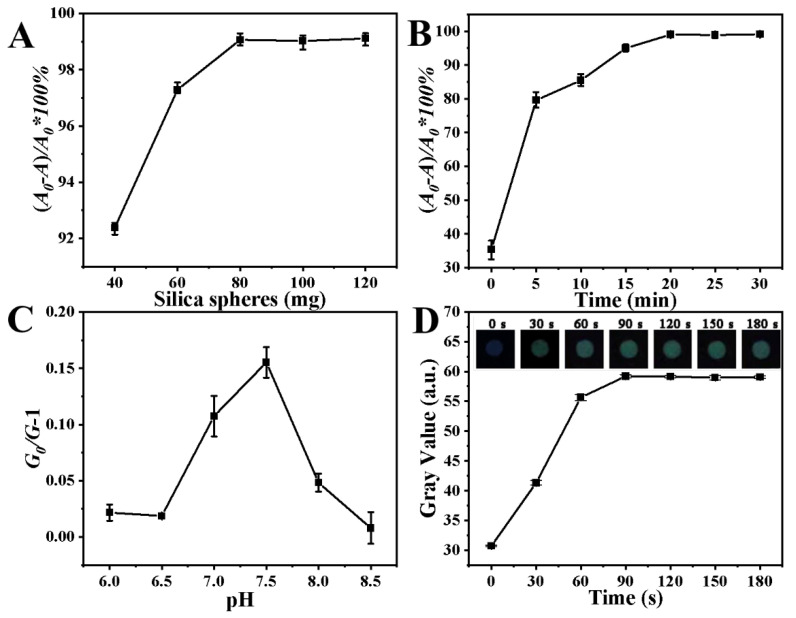
(**A**) Influence of the amount of silicon nanomaterials on the grafting rate of QDs. (**B**) Influence of the grafting time on the grafting rate of QDs. (**C**) Influence of pH on the quenching effects of QDs. (**D**) Fluorescence gray values of QDs–SiO_2_ versus time within 180 s, where the Hg^2+^ concentration was 50 μg/L, the excitation wavelength was 396 nm and the slit widths were 5 nm.

**Figure 4 sensors-23-03094-f004:**
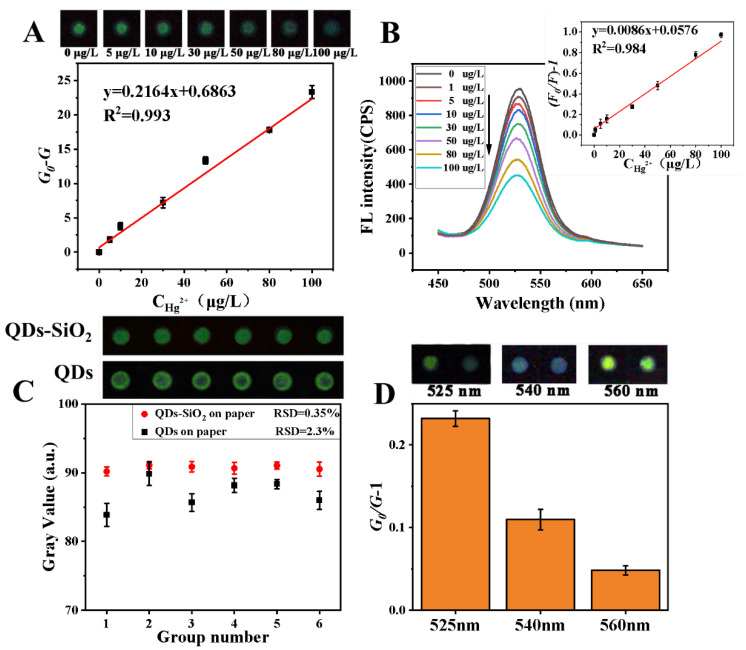
(**A**) Photograph of the color change of the quenching effect of Hg^2+^ on the paper-based QDs–SiO_2_ and its corresponding linear curve with linear equation. The concentrations of Hg^2+^ were 0, 5, 10, 30, 50, 80 and 100 μg/L. (**B**) Fluorescence spectral response diagrams of the quenching effect of Hg^2+^ on paper-based QDs–SiO_2_ and its corresponding linear curve with linear equation. The concentrations of Hg^2+^ were 0, 1, 5, 10, 30, 50, 80 and 100 μg/L, the excitation wavelength was 396 nm and the slit widths were 5 nm. (**C**) The gray values of six different sensing sites with QDs–SiO_2_ and QDs. (**D**) The influence of the different emission wavelengths of QDs on the quenching effects, where the Hg^2+^ concentration was 50 μg/L (n = 3).

**Table 1 sensors-23-03094-t001:** Determination of Hg^2+^ ions in actual samples using a paper-based chip (n = 5).

Water Samples	Spiked Concentration (μg/L)	Detection Concentration (μg/L)	Recovery and RSD (%)
	10	9.87	98.7 ± 4.3
**Seawater sample 1**	50	52.4	104.8 ± 2.8
	100	104.2	104.2 ± 2.4
	10	10.03	100.3 ± 3.2
**Seawater sample 2**	50	48.9	97.8 ± 5
	100	98.67	98.7 ± 2.2
	10	10.54	105.4 ± 4
**Seawater sample 3**	50	52.4	104.8 ± 3.2
	100	103.55	103.55 ± 3.5
	10	9.68	96.8 ± 4.4
**Lake water**	50	49.57	99.14 ± 3.5
	100	102.4	102.4 ± 1.3
	10	10.42	104.2 ± 2.7
**River water**	50	52.61	105.22 ± 2.8
	100	105.4	105.4 ± 2.2
	10	9.66	96.6 ± 3.1
**Tap water**	50	50.4	100.8 ± 2.3
	100	101.15	101.15 ± 2.9

## Data Availability

Not applicable.

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
