# Peer review of "A Simple and Effective Visual Fluorescent Sensing Paper-Based Chip for the Ultrasensitive Detection of Mercury Ions in Environmental Water"

_sensors, 2023, doi:10.3390/s23063094_

Round 1
Reviewer 1 Report
This work reported a novel, simple, and effective visual fluorescent sensing paper-based chip for the ultrasensitive detection of mercury ions in environmental water. The method is sensitive, low-cost, and effective. Nevertheless, I have several comments that the authors must discuss before the manuscript can be considered for publication. A minor revision is needed in the current situation.
Comments:
What is the pH value for the investigation of the effect of buffer systems on sense? Please give the relevant data.
The model of the smartphone used in the visual inspection is missing. In addition, the photographic condition should be described. Please give relevant information.
Please note that some formatting errors are to be checked. For example, the number 1 in the formula (G0/G)-1 should not be in italics.
Please correct some inappropriate expressions. In the section on selectivity, the authors state that the effect of interferents on sensing is within acceptable limits, which proves that QDs-SiO2 has good resistance to interference. “Good resistance to interference” should be changed to good selectivity.
Author Response
Reviewer 1
Comments and Suggestions for Authors
This work reported a novel, simple, and effective visual fluorescent sensing paper-based chip for the ultrasensitive detection of mercury ions in environmental water. The method is sensitive, low-cost, and effective. Nevertheless, I have several comments that the authors must discuss before the manuscript can be considered for publication. A minor revision is needed in the current situation.
1.What is the pH value for the investigation of the effect of buffer systems on sense? Please give the relevant data.
Response: Thanks for the reviewer’s insightful and helpful comments. Your suggestion is great appreciated. The related descriptions about the pH value for the investigation of the effect of buffer systems on sense has now been added to the revised manuscript. Please see page 6, lines 24-25 and the description of Figure S1. Thank you again for your helpful suggestions.
Changes:
We have investigated three buffer systems (PBS, Tris, and HEPES, pH 7.5) on the paper-based chip. (Page 6, lines 24-25)
All buffer system concentrations were 0.01M, pH were 7.5. The experimental conditions were room temperature. (The description of Figure S1)
- The model of the smartphone used in the visual inspection is missing. In addition, the photographic condition should be described. Please give relevant information.
Response: Thanks for the reviewer’s insightful and helpful comments. The related descriptions about the model of the smartphone and the photographic condition has now been added to the revised manuscript. Please see page 3, lines 26-27, and page 4, lines 16-22. Thank you again for your helpful suggestions.
Changes:
The model of the smartphone used in the visual inspection was HUAWEI VOG-AL10. (Page 3, lines 26-27)
The 5 μL sample contained mercury ions was added to the reaction area and waited for 90 s at room temperature, and the color signal of green fluorescence on paper-based chip was collected by smartphone camera in the UV light (365 nm) box, the UV lamp power is 6w. The smartphone was located 25 cm above the paper-based chip and kept fixed, flash was turned off and ensured that the image was in the center of the smartphone. Smartphone captured photos with auto-focus mode. (Page 4, lines 16-22)
- Please note that some formatting errors are to be checked. For example, the number 1 in the formula (G0/G)-1 should not be in italics.
Response: Thanks very much for the kind reminder. Your suggestion is great appreciated. I have corrected the formatting errors in the revised manuscript. Please see page 6, lines 25-26 and the description of Figure S1. Thank you again for your helpful suggestions.
Changes:
The effect of sensing can be indicated by the (G0/G)-1. (Page 6, lines 25-26)
The effect of fluorescence quenching can be fitted by the following equation:(G0/G)-1. (The description of Figure S1)
- Please correct some inappropriate expressions. In the section on selectivity, the authors state that the effect of interferents on sensing is within acceptable limits, which proves that QDs-SiO2 has good resistance to interference. “Good resistance to interference” should be changed to good selectivity.
Response: Thanks very much for the kind reminder. Your suggestion is great appreciated. I have corrected the inappropriate expression of the section on selectivity in the revised manuscript. Please see page 9, lines 21-22. Thank you again for your helpful suggestions.
Changes:
As shown in Figure S2, the impact of interferents on sensing is within acceptable limits, which proves that QDs-SiO2 has a good selectivity. (Page 9, lines 21-22)
Reviewer 2 Report
I think it is a very interesting work. But I have a few questions and suggestions which I think would improve the quality of this manuscript.
1. What is the pattern that was printed on the paper based chip? Please show the design in Figure 1.
2. As I understand the graph in Figure 4C shows categorical data on the x-axis. So it must be changed to the appropriate categories instead of the current x-axis.
3. Why is there this large difference in recoveries?
4. I think the conclusion needs to be expanded.
Author Response
Comments and Suggestions for Authors
I think it is a very interesting work. But I have a few questions and suggestions which I think would improve the quality of this manuscript.
- What is the pattern that was printed on the paper based chip? Please show the design in Figure 1.
Response: Thanks very much for the kind reminder. Your suggestion is very helpful. A schematic diagram of the printed pattern and enlargement of the paper chip has been added to the Figure 1. Thank you again for your helpful suggestions.
Changes:
Figure 1. Schematic diagram of the manufacturing and testing of paper-based chip.
- As I understand the graph in Figure 4C shows categorical data on the x-axis. So it must be changed to the appropriate categories instead of the current x-axis.
Response: Thanks very much for the kind reminder. Your suggestion is great appreciated. The x-axis for Figure 4C has been changed in the revised manuscript. I changed the x-axis to "Group number" and added categorical data as “QDs-SiO2 and QDs ” to the legend. Thank you again for your helpful suggestions.
Changes:
Figure 4. (A) Photograph of the color change of the quenching effect of Hg2+ on the paper-based QDs-SiO2 and its corresponding linear curve with linear equation. The concentration of Hg2+ was 0, 5, 10, 30, 50, 80, 100 μg/L. (B) Fluorescence spectral response diagrams of the quenching effect of Hg2+ on paper-based QDs-SiO2 and its corresponding linear curve with linear equation. The concentration of Hg2+ was 0, 1, 5, 10, 30, 50, 80, 100 μg/L, the excitation wavelength was 396 nm, and the slit widths were 5 nm. (C) Gray values of 6 different sensing sites with QDs-SiO2 and QDs. (D) Influence of the different emission wavelengths of QDs on the quenching effects, Hg2+ concentration was 50 μg/L (n = 3).
- Why is there this large difference in recoveries?
Response: Thanks for the reviewer’s insightful and helpful comments. Actually, the recovery is within the controllable and acceptable range. Generally, recoveries of analysis of environmental heavy ions in the range of 90-110% are acceptable, and this method gave a fully acceptable recoveries of 96.8-105.4%. So although there is difference in the recoveries, it can also prove that the performance of the paper-based chip is satisfactory in actual samples. In addition, there is the reason for the large differences in recoveries is as follow: There are content differences between water samples. For simplifying of testing process, we did not use sample pretreatment techniques to deal with environmental water sample, and we are unable to completely remove organic matters from the environmental water samples. These organic matters are possible to adsorb Hg2+ that is the reason why it maybe causes differences in recoveries. Thank you again for your helpful suggestions.
- I think the conclusion needs to be expanded.
Response: Thanks for the reviewer’s insightful and helpful comments. Your suggestion is great appreciated. The related expanded descriptions about the conclusion part have now been added to the revised manuscript. Please see page 11, lines 5-10. Thank you again for your helpful suggestions.
Changes:
After optimization of the reaction conditions, this method was able to achieve a limit of detection of 2.83 μg/L. Using a fluorescence spectrophotometer, the limit of detection could be reduced to 0.38 μg/L. The method was effective and had a good selectivity. We successfully achieved the trace spiking detection of seawater (from three regions), lake water, river water and tap water with recoveries in the range of 96.8-105.4% by this method. (Page 11, lines 5-10)

Round 2
Reviewer 2 Report
While the authors have made improvements to the original manuscript, the limitations of the study mentioned by the author in their response should also be specified in the manuscript. I am refering to the following response:
"Thanks for the reviewer’s insightful and helpful comments. Actually, the recovery is within the controllable and acceptable range. Generally, recoveries of analysis of environmental heavy ions in the range of 90-110% are acceptable, and this method gave a fully acceptable recoveries of 96.8-105.4%. So although there is difference in the recoveries, it can also prove that the performance of the paper-based chip is satisfactory in actual samples. In addition, there is the reason for the large differences in recoveries is as follow: There are content differences between water samples. For simplifying of testing process, we did not use sample pretreatment techniques to deal with environmental water sample, and we are unable to completely remove organic matters from the environmental water samples. These organic matters are possible to adsorb Hg2+ that is the reason why it maybe causes differences in recoveries. Thank you again for your helpful suggestions."
The authors should mention their inability to completely remove the organic matter from the environmental water samples and them causing absorption of additional Hg2+.
You didn't perform pretreatment
and other such shortcomings of the methodology
Author Response
Comments and Suggestions for Authors
While the authors have made improvements to the original manuscript, the limitations of the study mentioned by the author in their response should also be specified in the manuscript. I am refering to the following response:
"Thanks for the reviewer’s insightful and helpful comments. Actually, the recovery is within the controllable and acceptable range. Generally, recoveries of analysis of environmental heavy ions in the range of 90-110% are acceptable, and this method gave a fully acceptable recoveries of 96.8-105.4%. So although there is difference in the recoveries, it can also prove that the performance of the paper-based chip is satisfactory in actual samples. In addition, there is the reason for the large differences in recoveries is as follow: There are content differences between water samples. For simplifying of testing process, we did not use sample pretreatment techniques to deal with environmental water sample, and we are unable to completely remove organic matters from the environmental water samples. These organic matters are possible to adsorb Hg2+ that is the reason why it maybe causes differences in recoveries. Thank you again for your helpful suggestions."
The authors should mention their inability to completely remove the organic matter from the environmental water samples and them causing absorption of additional Hg2+.
You didn't perform pretreatmentand other such shortcomings of the methodology.
Response: Thanks for the reviewer’s insightful and helpful comments. Your suggestion is great appreciated. The related descriptions about the limitations of the study has now been added to the revised manuscript. Please see page 10, lines 19-25. Thank you again for your helpful suggestions.
Changes:
There are content differences between water samples. Organic matters in environmental water samples may adsorb Hg2+, resulting in reduced content of free Hg2+ in the water samples. For simplifying of testing process, we did not use sample pretreatment techniques to deal with environmental water samples, and we were unable to completely remove organic matters from the environmental water samples. However, in the future we will combine this method with pretreatment techniques and the effect of organic matters will be avoided.(Page 10, lines 19-25)
